# Pros and Cons for Automated Breast Ultrasound (ABUS): A Narrative Review

**DOI:** 10.3390/jpm11080703

**Published:** 2021-07-23

**Authors:** Ioana Boca (Bene), Anca Ileana Ciurea, Cristiana Augusta Ciortea, Sorin Marian Dudea

**Affiliations:** 1Department of Radiology, “Iuliu Hatieganu” University of Medicine and Pharmacy, 400012 Cluj-Napoca, Romania; ancaciurea@hotmail.com (A.I.C.); sdudea1@gmail.com (S.M.D.); 2Department of Radiology, Emergency County Hospital, 400006 Cluj-Napoca, Romania; cristianaciortea@yahoo.com

**Keywords:** breast cancer, diagnosis, ultrasound, automated breast ultrasound, mammography

## Abstract

Automated breast ultrasound (ABUS) is an ultrasound technique that tends to be increasingly used as a supplementary technique in the evaluation of patients with dense glandular breasts. Patients with dense breasts have an increased risk of developing breast cancer compared to patients with fatty breasts. Furthermore, for this group of patients, mammography has a low sensitivity in detecting breast cancers, especially if it is not associated with architectural distortion or calcifications. ABUS is a standardized examination with many advantages in both screening and diagnostic settings: it increases the detection rate of breast cancer, improves the workflow, and reduces the examination time. On the other hand, like any imaging technique, ABUS has disadvantages and even some limitations. Many disadvantages can be diminished by additional attention and training. Disadvantages regarding image acquisition are the inability to assess the axilla, the vascularization, and the elasticity of a lesion, while concerning the interpretation, the disadvantages are the artifacts due to poor positioning, lack of contact, motion or lesion related. This article reviews and discusses the indications, the advantages, and disadvantages of the method and also the sources of error in the ABUS examination.

## 1. Introduction

Screening mammography in women with dense breasts has reduced sensitivity. Dense glandular tissue is an independent risk factor in the development of breast cancer, the risk being 6–8 times higher than in women with fatty breasts. Therefore additional imaging modalities are required to improve cancer detection [1,2].

Automated breast ultrasound (ABUS) represents a new imaging technique approved by the Food and Drug Administration (FDA) in 2012 as a supplemental screening tool for women with heterogeneously and extremely dense breasts. Approval became possible after the U-Systems pivotal clinical retrospective multi-reader study. The study included 164 cases, 133 non-cancers and 31 biopsy-proven cancers. A total of 17 radiologists first interpreted the mammograms alone and then interpreted the combined mammography and ABUS. The authors found that ABUS could detect breast cancer with a clinically insignificant decrease in specificity compared to screening mammography alone (76.2% vs. 78.1%; *p* = 0.48) [3]. ABUS is a technique that separates the moment of image acquisition (made by the radiographer) from the moment of image interpretation, thus reducing the operator-dependence, as well as the time spent by the physician. In addition, coronal reconstructions bring new diagnostic information. Therefore, this technique was developed in order to standardize breast ultrasound and to eliminate some limitations of hand-held ultrasounds (HHUS), such as operator dependence and time of examination [4].

The aim of this paper is to review the advantages and drawbacks of ABUS and also to illustrate the main artifacts that could limit the diagnosis. In the present review, we performed a computerized search by using the PubMed database (www.ncbi.nlm.nih.gov/pubmed/, accessed on 30 April 2021), including articles listed up to 30 April 2021. The following search terms were used: automated breast ultrasound/ultrasonography, automated breast volume scanner/scanners/scanning, and automated whole breast ultrasound/ultrasonography artifacts. Only articles in English were included. Titles and abstracts of search results were examined. A total of 107 articles were considered suitable for full-text analysis. Articles regarding the use of ABUS in the screening or clinical setting were included.

## 2. Pros regarding ABUS

### 2.1. Screening

#### ABUS Associated with Full-Field Digital Mammography (FFDM)

SomoInsight is the most extensive study that assessed the diagnostic performance of ABUS in a screening scenario, including 15.318 asymptomatic women with dense breasts. By associating ABUS with FFDM, the increase in the detection rate was 1.9 per 1000 women, increasing sensitivity by 26.7%. The recall rate increased after combining the two methods (284.9 per 1000 women versus 150.2 per 1000 women in mammography alone) [5]. With increasing experience with ABUS, the recall rate decreases, and double reading could be the means to reduce this rate during early phase adoption [6]. ABUS associated with FFDM plays an important role, mainly in screening programs (Table 1).

### 2.2. Technique and Breast Cancer Detection

Considering that non-physician personnel acquire the images, ABUS has several advantages, such as decreased operator-dependence and high reproducibility. The image storage on a dedicated station allows multiplanar reconstructions and offers the possibility of double-reading and objective comparison with previous examinations [4]. 

The breasts are pendulous movable organs with different sizes, shapes, and densities. The receptor plate is not designed to fit all breasts, and the peripheral areas can be missed. In order to cover the entire breast, the technologists select the most suitable setting for each patient according to the breast size (A–D cups) and three to five views of each breast are acquired. A typical examination consists of three automated scans of each breast in the anterior-posterior, medial, and lateral views. In the case of large breasts, additional views of the superior and inferior parts of the breasts are required [10]. Breast size is a particular aspect of every woman and variable in the general population. Golatta et al. [11] found in their monocentric, exploratory, prospective study that included 983 patients that ≥14.8% had cup D breast size, which may have required additional acquisitions.

#### The Usefulness of the Coronal Plane

Since, in the coronal plane, the breast is seen as on the operating table, this plane is known as the surgical plane [12]. This view improves the assessment of the edges of the lesions. 

Benign lesions are frequently surrounded by a continuous hyperechoic rim, while malignant lesions often present a discontinuous hyperechoic rim [12,13]. In addition, in this plane, cancers are recognized as “black holes”, frequently associated with a retraction phenomenon represented by tumor infiltration and the desmoplastic reaction of the surrounding tissue against the malignancy, appearing as hyperechoic straight lines radiating from the surface of the mass (Figure 1) [14,15]. Zelst et al. [16] found a positive correlation between a proposed “spiculation and retraction phenomenon index” evaluated in the coronal plane and the likelihood of cancer. The retraction phenomenon is correlated statistically with smaller tumor size, lower histological grade, and positive status of estrogen and progesterone receptors. In contrast, the association hyperechoic rim-retraction phenomenon may relate to a better prognosis [17].

The coronal plane allows the reconstruction of the ductal system of the entire breast, which facilitates the detection of ductal dilatation associated with intraductal papillary lesions [6], or even ductal carcinoma in situ, by detecting intraluminal echoes in dilated lactiferous ducts [18]. 

In addition, ABUS showed promising results for the preoperative evaluation of tumor extension, being even superior to HHUS [19]. 

Wang et al. [20] found that the coronal plane can help predict the complete pathological response after two cycles of neoadjuvant chemotherapy with a sensitivity of 85.7–88.1% and specificity of 81.5–85.1%, respectively.

### 2.3. ABUS as a Reproducible Method

ABUS provides reproducible images for breast lesion location, size measurement, and characterization, which proved useful, especially in clinical situations requiring follow-up imaging. Chang et al. [21] studied 24 patients (24 cancers and 9 lesions) who underwent bilateral ABUS twice at an average interval of 1.3 days (before biopsy or surgery). Of the 33 lesions, only 31 were found at both readings. The readers assessed the clockface position, distance from the nipple, depth, size, and ultrasound characteristics. The interclass correlation coefficients (ICC) indicated an excellent reliability in terms of clockface position (ICC = 0.926; 95% CI: 0.986–0.997), distance from the nipple (ICC = 0.926; 95% CI: 0.840–0.913), and size (ICC = 0.980; 95% CI: 0.955–0.991). They found low reliability for depth (ICC = 0.342 (95%; CI: −0.053 to 0.645), while 29 lesions of the 31 were classified as identical or similar. Scanning pressure variation may influence the depth, and it also may affect lesion characteristics. However, the authors, in a recently updated system adjusted the compression pressure in five steps with a maximum pressure of 25 lbs.

The detection rate of breast cancer per 1000 women screened increased by 2.4 [9], 3.9 [7], and even 7.7 [8] for ABUS associated with FFDM compared to FFDM alone. The recall rate per 1000 women screened in the retrospective reader study conducted by Wilczek et al. [9] was 13.8 (95% CI: 9.0, 19.8) for FFDM alone and 22.8 for combined FFDM and ABUS (95% CI: 16.2, 30.0). 

### 2.4. ABUS Associated with Digital Breast Tomosynthesis (DBT)

DBT represents a new imaging technique able to overcome the limits of FFDM induced by the tissue superimposition effect and achieve better lesion conspicuity [22]. 

Girometti et al. [23] found in their retrospective clinical study a similar diagnostic accuracy of ABUS + DBT and magnetic resonance imaging (MRI) in preoperative breast cancer staging. A total of 71 women were included in the study with 160 lesions (52 benign and 108 malignant). They obtained a lower sensitivity and positive predictive value for supplemental disease by ABUS + DBT compared with MRI (Table 2).

The most reliable technique for breast cancer staging is MRI. However, when MRI is not feasible or available, the association of ABUS and DBT might represent an accurate alternative to FFDM or HHUS.

### 2.5. ABUS Compared with HHUS

Hellgren et al. [24] compared in their prospective multi-reader clinical study the sensitivity and specificity of ABUS and HHUS for breast cancer detection in women recalled after mammography screening. They included 113 patients who underwent mammographic screening, and for whom a suspicious finding was reported and subsequently underwent HHUS and an additional ABUS exam. The methods evaluated each breast and each lesion separately by classifying them: breasts with mammography suspicion of malignancy and breasts with negative mammography. A total of 26 cancers were detected in 25 of the women. In breasts with mammography suspicion of malignancy, sensitivity was 88% for both ABUS and HHUS, and specificity was 93.5% for HHUS and 89.2% for ABUS. Sensitivity was 100% for the two methods regarding breasts with negative mammography, and specificity was 100% for HHUS and 94.1% for ABUS. Therefore, ABUS seems suitable to replace HHUS in women recalled for a suspicious finding on screening mammography. An important aspect that would be relevant to mention is the fact that HHUS gives valuable information and allows better scanning of the peripheral region. Furthermore, the experience and the knowledge of the clinician increase the quality of the diagnosis.

The rate of breast cancer detection using ABUS increases with the size of the lesion. Berg et al. [25] reported a 12% increase in the detection of breast lesions per millimeter with an increase in their diameter between 3 and 13 mm. For breast masses between 3.1 and 5 mm, the detection rate was 44%, and for those over 11 mm, the detection rate reached 97%. Some studies have obtained a significantly higher detection rate in ABUS compared to HHUS. Wang et al. [26] reported detection rates of HHUS and ABUS were 95.8% (158/165) and 97.6% (161/165), respectively. Furthermore, the three cases missed by US were found by ABVS, which illustrates a key advantage of an ABUS system for standardized, reproducible, and bilateral whole-breast imaging. Zhang et al. [27] obtained a breast cancer detection rate of 89.9% in ABUS versus 60.6% in HHUS, while Xiao et al. [28] included 300 patients and found a detection rate using ABUS of 100% and 78.2% using HHUS. 

Table 3 shows the comparison in terms of sensitivity and specificity of the ABUS and HHUS in several studies, including a significant number of women.

### 2.6. ABUS Compared with MRI

Schmachtenberg et al. [31] reported, in their retrospective clinical study, a good correlation between ABUS and MRI (k = 0.83) for lesion measurement. 

Girometti et al. [32] compared ABUS with HHUS as second-look methods for evaluating 131 patients who underwent MRI. The indications were high risk of developing breast cancer, B3 type lesions, and evaluation of the response to neoadjuvant chemotherapy. The results of the two methods were comparable in terms of detection rate (69.3% for ABUS versus 71.5% for HHUS) with an almost perfect agreement (k = 0.94). 

Chae et al. [33] evaluated the performance of second-look ABUS and second-look HHUS in the detection of additional suspicious lesions in 58 patients who underwent preoperative breast MRI. Second-look ABUS detected 70 of the 80 additionally detected lesions, and HHUS detected 65 of the 80 lesions, while 10% of the lesions detected at ABUS were unapparent at HHUS.

### 2.7. ABUS as a Time-Saving Method

Compared to HHUS, which is performed by physicians and involves 20 min per patient, the time the radiologist involves in ABUS is only related to the evaluation of the images because the technician makes the acquisition. Several studies analyzed the time required by the physician for reporting an ABUS examination. Skaane et al. [14] reported a mean interpretation time of around 9 min/patient (around 4 min/breast for a normal examination and 5 min/breast in cases with suspicious findings). Chae et al. [34] reported a mean interpretation time of 3.83 ± 1.71 min for the coronal plane and 5.57 ± 2.21 min for the transverse plane, while Huppe et al. [35] reported an average of more than 3 min for all three readers involved in the reading process. This study replicated a screening scenario that predominantly included normal examinations compared to a diagnostic population in the other studies, which would explain the time difference in interpretation. Due to this advantage, ABUS can be used as an alternative tool in screening women with dense breasts to improve the workflow.

### 2.8. ABUS as a Money-Saving Method

Foglia et al. [36] analyzed the budget impact of using ABUS in breast cancer screening. As screening pathways, they compared the AS-IS scenario (involving mammography, with the further examination if anything suspicious is detected), IDEAL scenario (mammography + HHUS), and TO-BE scenario (mammography + ABUS). They found that the last scenario could lead to an economic saving equal to or greater than 54 million euros for the Italian National Healthcare Service. Compared to the AS-IS scenario and IDEAL scenario, the TO-BE scenario has a probability of 77% and 80%, respectively, to absorb fewer resources. To the best of our knowledge, this is the only study regarding the economic impact of ABUS utilization in screening. The limitations of the study are that data related to ultrasound recall rates, process mapping of treatment, and care pathway of women affected by cancer were retrieved from real-world data, and the lack of recent information related to the rate of annual invites to participate in screening programs.

### 2.9. Computer-Aided Detection (CAD) Adapted to ABUS

Recently developed CAD (computer-aided detection) software for ABUS (QVCAD™, QView Medical, Los Altos, CA, USA) received the Food and Drug Administration (FDA) approval and offers promising performance regarding cancer detection. In a retrospective observer performance study, it induced an improvement of the mean interpretation time with an average of 33% among 18 radiologists without influencing the diagnostic accuracy. CAD was tested in a standardized setup; each reader interpreted each case twice, once without the CAD system and once with it, separated by 4 weeks. Without QVCAD, the mean interpretation time was around 3.33 min per case, and it decreased to 2.24 min per case after it was applied [37]. 

## 3. Cons Regarding ABUS

### 3.1. Technique and Artifacts

#### 3.1.1. Limitations Related to the Technique

Lesions might be missed on ABUS if they have a peripheral location. This technical drawback reduces the diagnostic performance of the method compared to HHUS, especially in large breasts, and could represent a cause for the misdiagnosis of cancer. The radiographer should be aware of this aspect and scan the entire breast by obtaining supplemental acquisitions on the superior and inferior parts of the breasts [21]. 

The main limitation of ABUS is its inability to assess the axilla, the absence of information regarding the lymph node status, the vascularization, and the elasticity of a lesion [38]. There is, however, progress in this regard, Hendriks et al. [39] proposed a 3D ultrasound quasi-static elastography method on an ABUS-like device in a preclinical environment. Wang et al. [40] tested a 3D motion-tracking system that apparently can effectively follow the dislocation of the lesions in the three planes, thus providing information related to their elasticity. Furthermore, invasive procedures performed under ABUS guidance are an important limitation; therefore lesions detected with ABUS and requiring further assessment need to be reevaluated with HHUS.

#### 3.1.2. Artifacts

##### Artifacts Induced by the Use of Ultrasound Gel

ABUS does not use classic ultrasound gel because of the artifact inducing tiny gas bubbles it may contain. To avoid these artifacts, ABUS requires the use of a gel specially developed for this purpose, which has the consistency of a homogeneous lotion (Figure 2).

##### Air Interposition 

If the lotion used is not evenly spread and is missing in a region, the air gets interposed between the transducer and the skin, the sound waves are reflected by the air between the transducer membrane and skin, inducing shadowing and making the visualization of the underlying glandular tissue impossible (Figure 3) [41]. 

##### Insufficient Compression 

In the case that the transducer is not evenly and sufficiently compressed, the air becomes interposed at the edges of the acquired image, hampering the analysis of the glandular parenchyma in the periphery of the image. Insufficient compression may also cause artifacts induced by Cooper ligaments, an artifact also found in the case of insufficient compression at HHUS (Figure 4). In order to reduce this artifact, the radiographer should perform an adequate compression.

##### Probe Motion Artifacts

If the transducer sliding on the skin surface is not uniform due to insufficient lotion, the “edge” artifacts appear. They are seen as hypoechogenicity located on the side of the axial scan field, oriented vertically to the skin’s surface not corresponding to a scar, and which does not persist on dedicated lateral acquisition [42]. 

##### Breathing Artifacts

Rapid or deep breathing induces oscillating movement of the chest wall, causing the appearance of “wave-like” artifacts [43]. 

##### Skip Artifact 

When the transducer quickly slides over a firm and superficial lesion (cyst, fibroadenoma), it creates a linear artifact, which is observed in the coronal plane and sagittal plane. It appears as a horizontal line located superior to the lesion [44] (Figure 5).

##### Nipple Artifact

The retro areolar region is difficult to assess due to the shadowing artifact induced by the nipple that appears as hypoechoic columns extending in the anterior-posterior direction behind the nipple. This artifact can be caused by an imperfect adjustment of irregular nipple surface, and if an abnormality is suspected, the patient is recalled for rescanning or for an HHUS (Figure 6) [44]. 

##### Attenuation Areas 

Post-traumatic or post-therapeutic edema, or edema due to infectious or carcinomatous mastitis, causes the appearance of diffuse attenuation areas inside the glandular parenchyma (Figure 7).

##### White Wall Sign

This artifact is observed with transonic masses (cysts) or some solid masses as in HHUS. The white-wall sign presents as an echogenic wall in the coronal view and corresponds to the acoustic enhancement on HHUS. It appears posterior to the lesion due to less attenuated ultrasounds within the lesion. It can help interpretation, but it is not specific to benign masses due to the fact that high-grade carcinomas can also present this artifact (Figure 8) [45,46]. 

### 3.2. False-Positive Results

The pathologies that can represent causes of false-positive results are adenosis, intraductal papilloma, fibroadenoma, or mastitis [12]. 

ABUS images are sectional and static images. This technical peculiarity opens the possibility to obtain false-positive results in complicated cysts mimicking complex cystic lesions. In these situations, HHUS may obviate in real-time the fact that the intracystic appearance is produced only by floating echoes (Figure 9).

### 3.3. False-Negative Results

Small lesions, circumscribed edges (as in medullary carcinoma, phyllodes tumors or invasive solid papillary carcinoma), or peripheral localization of the mass may be sources of false-negative results [12]. 

Removal or significant reduction of artifacts and lowering false-positive or negative results can be obtained through rigorous training of radiographers to respect the patient’s positioning, appropriate transducer-skin contact, adequate lotion usage, and complete scanning of glandular tissue in the peripheral regions. In addition, physicians should have information related to the history of patients and clinical data or even previous examinations for comparison.

Figure 10 illustrates benign lesions detected using ABUS.

## 4. Conclusions

ABUS is a new imaging technique with its advantages and disadvantages. Many disadvantages can be diminished by additional attention and training, both for image acquisition and interpretation. ABUS is a method promising to improve and ease breast cancer screening.

## Figures and Tables

**Figure 1 jpm-11-00703-f001:**
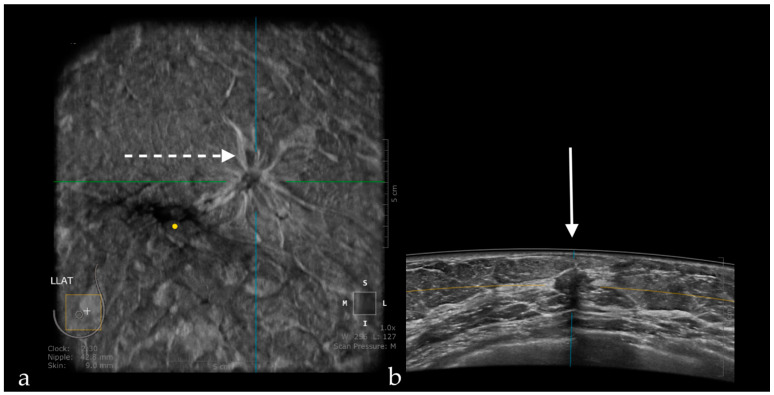
A case of invasive ductal carcinoma in coronal (**a**) and axial (**b**) planes. The lesion presents as a hypoechoic, non-circumscribed (spiculated) mass (white arrow). In the coronal plane, the spiculations due to the desmoplastic reaction are more obvious than in the axial plane, presenting as hyperechoic, straight lines radiating from the surface of the mass (white dotted arrow).

**Figure 2 jpm-11-00703-f002:**
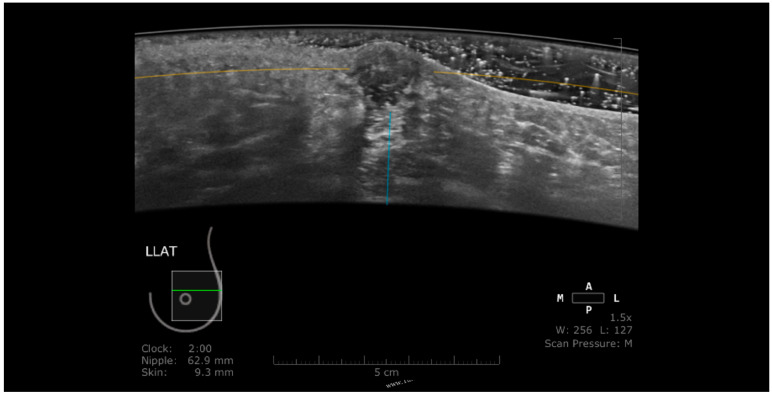
Permeation nodule appearing as a hypoechoic intradermal lesion. A significant amount of common ultrasound gel was applied to visualize the lesion. The gel contains small air bubbles appearing as multiple hyperechoic dots with comet-tail artifacts.

**Figure 3 jpm-11-00703-f003:**
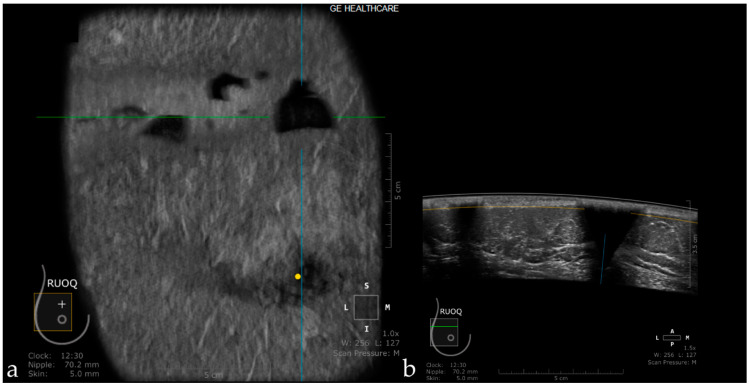
Coronal (**a**) and axial planes (**b**) in a case with multiple air interpositions between the skin and the transducer. The air appears as hypoechoic images with posterior shadowing.

**Figure 4 jpm-11-00703-f004:**
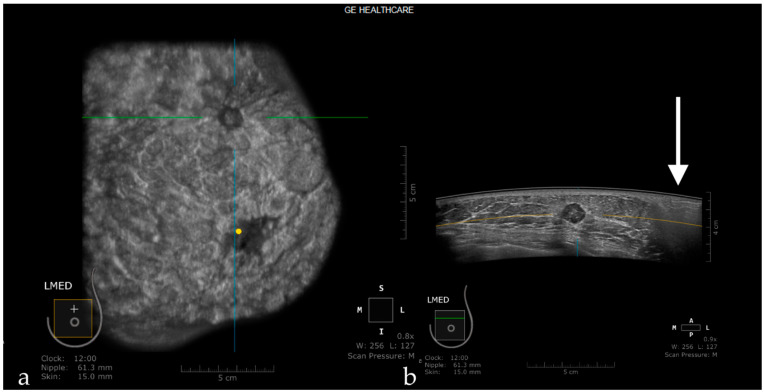
The coronal plane (**a**) reveals a hypoechoic suspicious mass, also visible on the axial acquisition (**b**). Due to an inappropriate compression, the glandular tissue below the edge of the transducer cannot be appreciated (white arrow).

**Figure 5 jpm-11-00703-f005:**
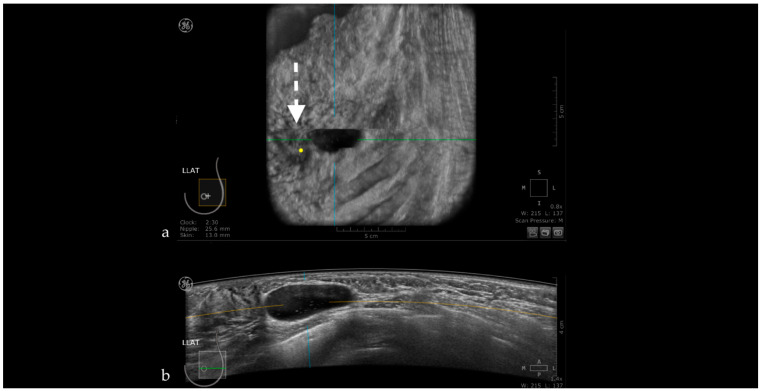
The case of a bulging cyst visible on both coronal (**a**) and axial planes (**b**) at 2.30 o’clock, 25.6 mm from the nipple (yellow dot). Due to the superficial location of the lesion, the transducer slides over its edge producing the skip artifact seen on the coronal plane (white dotted arrow).

**Figure 6 jpm-11-00703-f006:**
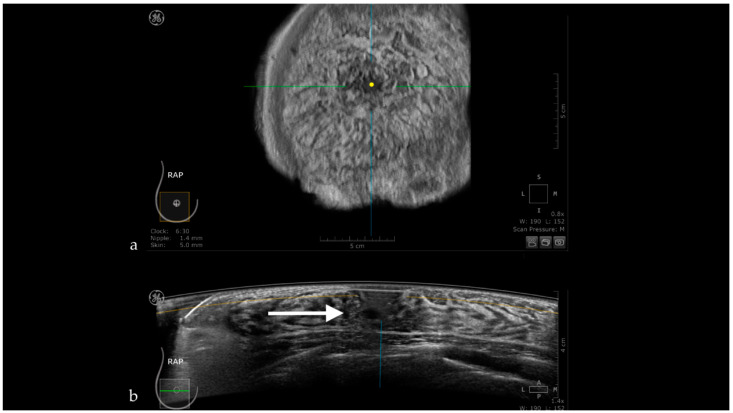
Hypoechoic columns (white arrow—**b**) behind the nipple (yellow dot—**a**) interfering with the evaluation of the breast tissue: nipple artifact.

**Figure 7 jpm-11-00703-f007:**
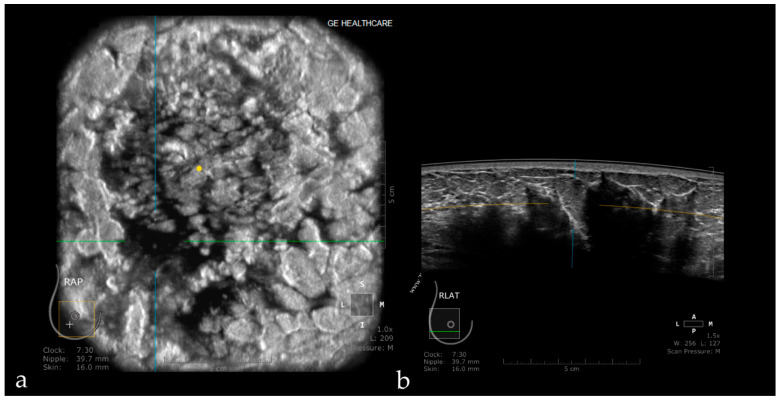
Diffuse attenuation areas inside the glandular parenchyma due to post-therapeutic edema, visible on the coronal (**a**) and axial planes (**b**).

**Figure 8 jpm-11-00703-f008:**
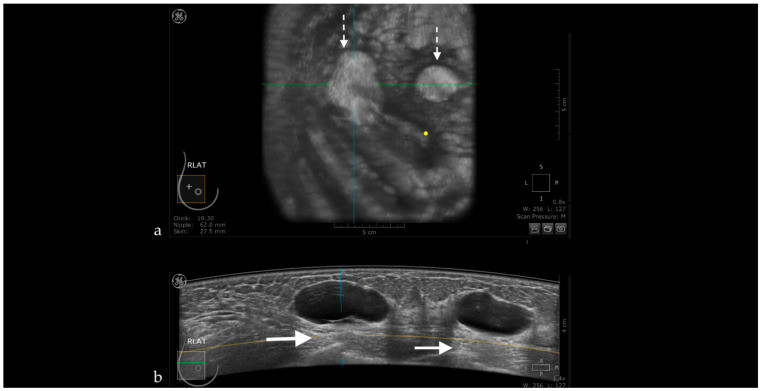
Hyperechoic oval images seen on the coronal plane (white dotted arrows—**a**) corresponding to the posterior enhancement observed on the axial plane behind the breast cysts (white arrows—**b**).

**Figure 9 jpm-11-00703-f009:**
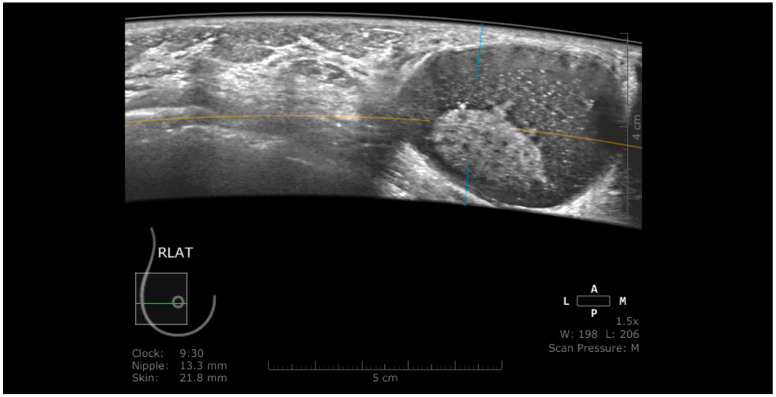
Complicated cyst mimicking a complex cystic lesion using ABUS. Using HHUS, the intracystic component was proved to be only abundant, floating echoes.

**Figure 10 jpm-11-00703-f010:**
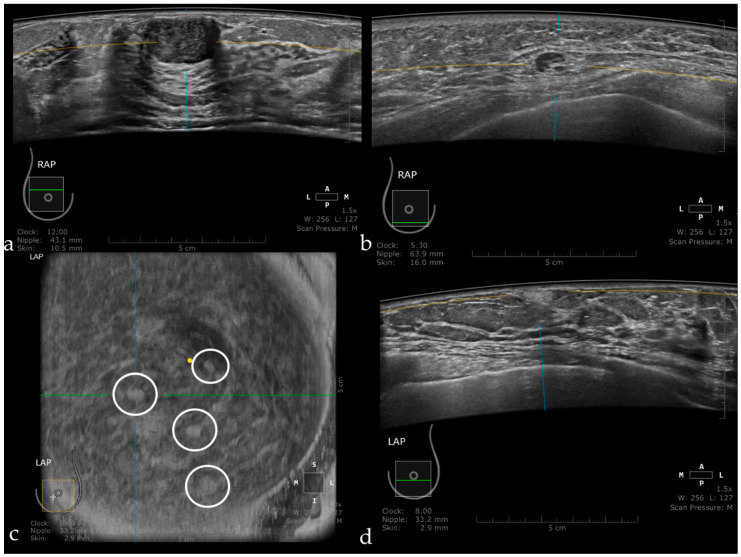
Fibroadenoma appearing as a hypoechoic, circumscribed mass, with the long axis parallel to the skin (**a**). Intramammary lymph node seen as a circumscribed mass, with a hyperechoic center representing the hilum and a thin cortex in the periphery (**b**). Multiple angiolipomas observed on the coronal plane (inside the white circles) (**c**), and also on axial acquisition as a hyperechoic, homogeneous, circumscribed mass (**d**).

**Table 1 jpm-11-00703-t001:** The diagnostic value of FFDM versus FFDM + ABUS in screening programs.

Author	Year of Publication	Number of Patients	Sensitivity (%)		Specificity (%)	
			FFDM	FFDM + ABUS	FFDM	FFDM + ABUS
Brem et al. (SomoInsigh) [5]	2014	15,318	73.2	100	85.4	72
Kelly et al. [7]	2010	4419	40	81	95.15	98.7
Giuliano et al. [8]	2012	3418	76	96.7	99.70	98.2
Wilczek et al. [9]	2016	1668	63.6	100	99	98.4

**Table 2 jpm-11-00703-t002:** Comparison between ABUS + DBT versus MRI in preoperative staging of breast cancer.

	ABUS + DBT % (95% CI)	MRI % (95% CI)
Sensitivity	76.5 (58.8–89.3)	91.7 (84.6–96.1)
Positive predictive value	78.8 (61–91)	93.4 (86.9–97.3)
Diagnostic accuracy	90 (84.3–94.2)	93.8 (88–97)

**Table 3 jpm-11-00703-t003:** Sensitivity and specificity of ABUS versus HHUS regarding breast cancer detection.

Author	Year of Publication	Number of Participants	Sensitivity (%)		Specificity (%)	
			ABUS	HHUS	ABUS	HHUS
Wang et al. [12]	2012	213	95.3	90.6	80.5	82.5
Wang et al. [26]	2012	155	96.1	93.2	91.9	88.7
Chen et al. [13]	2013	175	92.5	88.1	86.2	87.5
Jen et al. [29]	2016	173	88	95.7	76.2	49.4
Niu et al. [30]	2019	398	92.23	82.52	77.62	80.24

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
