# Peer review of "Pros and Cons for Automated Breast Ultrasound (ABUS): A Narrative Review"

_jpm, 2021, doi:10.3390/jpm11080703_

Round 1

Reviewer 1 Report

This was an intersting manuscript to read regarding a novel technique that may have advantages compared to the present standardized approach to breast cancer. However, the results of this review was not obtained in a standardized way, in fact no information is giving about how the cited papers were chosen for this review, and the information is presented without sufficient references.

Major / general concerns:

  1. Please revise the aim. This study should more clearly state how the content of the manuscript was chosen, please provide a systematic search string or meticulously address how the references of this manuscript was chosen. All, relevant information provided in manuscript should have a reference.

Otherwise, add to title that this is a narrative review.

Only use common abbreviations.

For all studies mentioned please address in which set-up the data was achieved, clinical and blinded or experimental and non-clinical set-up? This is of great importance, since the validity of the results are dependent on how standardized they were obtained.

In general, too many results of studies are mentioned without evaluation of the validity of the referenced studies. This is especially of concern, since the study has not been performed in a standardized way with a systematic search string in multiple databases.

Does ABUS apply to all size and shapes of breasts? please address.

Very few references are provided in artefacts section of manuscript, please consider revising this section. Otherwise omit, and address in detail the results provided in the first half of manuscript.

Specific comments:

  1. Write sensitivity instead of sensibility.

31-34. Please address how many patients were examined in the cited article and if the study was performed in a clinical, standardized and blinded setting. Otherwise, limitations of the results of the study should be addressed.

83-84. Do not abbreviatie sensitivity and specificity.

101-102. Please, elaborate. What was the recall rate in these studies.

  1. Do not abbreviate PPV.
  2. Do not write “for the first category”, instead write mammography with suspicion of malignancy and the same for “the second category”.
  3. Omit information in parenthesis or add in additional sentence.

175 and 177. Only use abbreviations when they are explained beforehand and if they are commonly accepted in the general population.

  1. Provide abbreviations of CAD, i.e. computer aided diagnosis. Please write in what circumstances the CAD was tested, in a standardized setup??? Otherwise consider to omit.

198-203. This section is far to general. Please provide additional information or consider to omit.

Author Response

Response to Reviewer 1 Comments

Point 1: Please revise the aim. This study should more clearly state how the content of the manuscript was chosen, please provide a systematic search string or meticulously address how the references of this manuscript was chosen. All, relevant information provided in manuscript should have a reference.

Otherwise, add to title that this is a narrative review.

Response 1: As a response to Point 1, we modified in the body text the following statements

“The aim of this paper is to review the advantages and drawbacks of ABUS and also to illustrate the main artifacts, which could limit the diagnosis. In the present review, we performed a computerized search by using the PubMed database (www.ncbi.nlm.nih.

gov/pubmed/), including articles listed up to 30 April 2021. The following search terms were used: automated breast ultrasound/ultrasonography, automated breast volume scanner/scanners/scanning and automated whole breast ultrasound/ultrasonography, artifacts. Only articles in English were included. Titles and abstracts of search results were examined. 107 articles were considered suitable for full-text analysis.Articles regarding the use of ABUS in the screening or clinical setting were included.“(Lines 46-55)

We also added to the title :” a narrative review” Lines 2-3

Point 2:For all studies mentioned please address in which set-up the data was achieved, clinical and blinded or experimental and non-clinical set-up? This is of great importance, since the validity of the results are dependent on how standardized they were obtained.

Response 2:As a response to Point 3: For the studies cited in the present article we have indicated their type (in which set-up the data were achieved) according to the mentions found in their Material and Method.

Point 3: Does ABUS apply to all size and shapes of breasts? please address.

Response 3:As a response to Point 3 we added the following statement: “The breasts are pendulous movable organs with different size, shape and density.The receptor plate is not designed to fit to all breasts and the peripheral areas can be missed. In order to cover the entire breast, the technologistsselect the most suitable setting for each patient according to the breast size (A-D cups) and three to five views of each breast are acquired. Atypical examination consists of three automated scans of each breast in the anterior-posterior, medial and lateral views. In case of large breasts additional views of the superior and inferior parts of the breasts are required [10]”.Lines (80-86)

Point 4:Very few references are provided in artefacts section of manuscript, please consider revising this section. Otherwise omit, and address in detail the results provided in the first half of manuscript.

Response 4:As a response to Point 4, we revised the Artifacts and we modified the following statements:

Air interposition

If the lotion used is not evenly spread and is missing in a region, the air gets interposed between the transducer and the skin, the sound waves are reflected by the air between transducer membrane and skin inducing shadowing and making the visualization of the underlying glandular tissue impossible (Figure 3) [40].(Lines 310-313)

Probe motion artifacts

If the transducer sliding on the skin surface is not uniform due to insufficient lotion, the "edge" artifacts appear. They are seen as hypoechogenicity located on the side of the axial scan field, oriented vertically to the skin's surface not corresponding to a scar and which does not persist on dedicated lateral acquisition [41].(Lines 335-338)

Breathing artifacts

Rapid or deep breathing induces oscillating movement of the chest wall, causing the appearance of "wave-like" artifacts [42].  (Lines 341-342)

Skip Artifact

When the transducer quickly slides over a firm and superficial lesion (cyst, fibroadenoma), it creates a linear artifact, which is observed in the coronal plane and sagittal plane. On the coronal plan it appears as a hyperechoic horizontal line located superior to the lesion [43] (Figure 5). (Lines 345-348)

Nipple artifact

The retro areolar region is difficult to assess due to the shadowing artifact induced by the nipple that appears as hypoechoic columns extending in anterior-posterior direction behind the nipple. This artifact can be caused by an imperfect adjustment of irregular nipple surface and if an abnormality is suspected the patient is recalled for rescanning or for a HHUS[43]  (Lines 358-362)

White wall sign         

This artifact is observed with transonic masses (cysts) or some solid masses as in HHUS (Figure 8). The white-wall sign presents as an echogenic wall in the coronal view and corresponds to the acoustic enhancement on HHUS. It appears posterior to the lesion due to less attenuated ultrasounds within the lesion. It can help interpretation, but it is not specific to benign masses due to the fact that high grade carcinomas can also present this artifact (Figure 8) [44-45] (Lines 383-389)

Point 5: 25 Write sensitivity instead of sensibility.

Response 5:As a response to Point 5, we modified in the body text the word sensibility with sensitivity

Point 6: 31-34. Please address how many patients were examined in the cited article and if the study was performed in a clinical, standardized and blinded setting. Otherwise, limitations of the results of the study should be addressed.

Response 6:As a response to Point 6, we modified in the body text the following statements:

Approval became possible after the U-Systems pivotal clinical retrospective multi-reader study. The study included 164 cases (133 non-cancers and 31 biopsy-proven cancers). 17 radiologists first interpreted the mammography images alone, and then interpreted the combined mammography+ABUS. The authors fund that ABUS could detect breast cancer with a clinically insignificant decrease in specificity compared to screening mammography alone (76.2% versus 78.1%; p=0.48). (Lines 33-39)

Point 7: 83-84. Do not abbreviatie sensitivity and specificity.

Response 7:As a response to Point 7, we deleted the abbreviations found in the body text

Point 8: 101-102. Please, elaborate. What was the recall rate in these studies.

Response 8:As a response to Point 8 we would like to mention that the recall rate was mentioned just in the first study [8]; The detection rate of breast cancer per 1000 women screen increased with 2.4 (8), 3.9 [6], and even 7.7 [7]for ABUS associated with FFDM compared to FFDM alone. The recall rate per 1000 women screened in the study conducted by Wilczek et al. [8]was 13.8 (95% CI: 9.0, 19.8) for FFDM alone and 22.8 for combined FFDM and ABUS (95% CI: 16.2, 30.0).(Lines 137-141)

Point 9: 110 Do not abbreviate PPV.

Response 9:As a response to Point 9, we deleted the abbreviations found in the body text (Line 150)

Point 10: 111 Do not write “for the first category”, instead write mammography with suspicion of malignancy and the same for “the second category”.

Response 10:As a response to Point 10, we modified in the body text the following statements: In breasts with mammography suspicion of malignancy, sensitivity was 88% for both ABUS and HHUS, and specificity was 93.5% for HHUS and 89.2% for ABUS. Sensitivity was 100% for the two methods regarding breasts with negative mammography, and specificity was 100% for HHUS and 94.1% for ABUS. (Lines 166-169)

Point 11: 112 Omit information in parenthesis or add in additional sentence.

Response 11: As a response to Point 11, we modified in the body text the following statements:  Girometti et al. [31] compared ABUS with HHUS as second-look methods for evaluating 131 patients who underwent MRI. The indications were high risk of developing breast cancer, B3 type lesions and evaluation of the response to neoadjuvant chemotherapy.(Lines 210-211)

Point 12: 175 and 177. Only use abbreviations when they are explained beforehand and if they are commonly accepted in the general population.

Response 12:As a response to Point 12, in the cited article we did not find the explanation of the words AS-IS, TO BE and IDEAL, so we deduced that they are not abbreviations, but more likely the name of the three scenarios.

Point 13: 183 Provide abbreviations of CAD, i.e. computer aided diagnosis. Please write in what circumstances the CAD was tested, in a standardized setup??? Otherwise consider to omit.

Response 13:As a response to Point 13, we modified in the body text the following statements: Recently developed CAD (computer-aided detection)software for ABUS (QVCAD™, QView Medical) received the Food and Drug Administration (FDA) approval and offers promising performance regarding cancer detection. In a retrospective observer performance studyit induced an improvement of the mean interpretation time with an average of 33% among 18 radiologists without influencing the diagnostic accuracy. CAD was tested in a standardized setup,each reader interpreted each case twice, once without the CAD system and once with it, separated by 4 weeks.(Lines 262-268)

Point 14:198-203. This section is far to general. Please provide additional information or consider to omit.

Response 14:As a response to Point 14, we modified in the body text the following statements:

The radiographer should be aware of this aspect and scan the entire breast, by obtaining supplemental acquisitions on the superior and inferior parts on the breasts [20].

The main limitation of ABUS is its inability to assess the axilla, the absence of information regarding the lymph node status, the vascularization and the elasticity of a lesion [37]. There is, however, progress in this regard, Hendriks et al. [38] proposed a 3D ultrasound quasi-static elastography method on an ABUS-like device in a preclinical environment. Wang et al. [39] tested a 3D motion-tracking system that apparently can effectively follow the dislocation of the lesions in the three planes, thus providing information related to their elasticity. Furthermore, invasive procedures performed under ABUS guidance are an important limitation. Therefore lesions detected at ABUS and requiring further assessment need to be reevaluate with HHUS. (Lines 283-293)

Dear reviewer,we would like to thank you for taking the time to carefully evaluate our article. Thank you for your suggestions, appreciation and for your positive response.

Best regards,

Ioana Boca (Bene)

Reviewer 2 Report

The paper submitted by Boca et al. provide important information of the pros and cons for ABUS. The review is well arranged,nonetheless, some changes must be considered before acceptance. These changes are listed below.  

-In the  abstract section, please include the cons.

-Since it is a review, the introduction section is too short, please extend.

-Line 33-4, this sentence is unclear, please clarify.

-Please add pics showing benign lesions. In addition, add arrows (Figure 1).

-Line 98-9, Scanning presure variation, add reference.

-Line 101-102, please redone this sentence since it is unclear.

-Line 128-129, it is important to highlight that HHUS examination will give important information and allow better scanning of the peripheral region.In addition, what about the experience and knowledge of the clinician?

-Line138-139, this statement is unclear  for me, Please explain the reason of this asseveration.

-Line 144, this section is confuse since the title do not fit with the information given. Please provide information about this comparison.

-Line 226-228: please redone this sentence.

-Line 247-249: in Figure 5, the skip artifact is unclear to me, Please check. 

-Posterior enhancement and nipple artifact sections are too short. please extend.

Author Response

Response to Reviewer 2 Comments

The paper submitted by Boca et al. provide important information of the pros and cons for ABUS. The review is well arranged,nonetheless, some changes must be considered before acceptance. These changes are listed below.  

Point 1: -In the abstract section, please include the cons.

Response 1:As a response to Point 1, we modified in abstract the following statement: “Many disadvantages can be diminished by additional attention and training, both for image acquisition (inability to assess the axilla, the vascularization and the elasticity of a lesion) and interpretation (artifacts due to poor positioning, lack of contact, motion or lesion related).” (Lines 18-21)

Point 2:-Since it is a review, the introduction section is too short, please extend.

Response 2:As a response to Point 2, we added in introduction section the following statements:

Approval became possible after the U-Systems pivotal clinical retrospective multi-reader study. The study included 164 cases (133 non-cancers and 31 biopsy-proven cancers). 17 radiologists first interpreted the mammograms alone, and then interpreted the combined mammography+ABUS. The authors found that ABUS could detect breast cancer with a clinically insignificant decrease in specificity compared to screening mammography alone (76.2% versus 78.1%; p=0.48) [3]. ABUS is a technique that separates the moment of image acquisition (made by the radiographer) from the moment of image interpretation, thus reducing the operator-dependence, as well as the time spent by the doctor. In addition, coronal reconstructions bring new diagnostic information. Therefore this technique was developed in order to standardize breast ultrasound and to eliminate some limitations of hand-held ultrasound (HHUS) such as operator-dependence and time of examination [4].

The aim of this paper is to review the advantages and drawbacks of ABUS and also to illustrate the main artifacts, which could limit the diagnosis. In the present review, we performed a computerized search by using the PubMed database (www.ncbi.nlm.nih.gov/pubmed/), including articles listed up to 30 April 2021. The following search terms were used: automated breast ultrasound/ultrasonography, automated breast volume scanner/scanners/scanning and automated whole breast ultrasound/ultrasonography, artifacts. Only articles in English were included. Titles and abstracts of search results were examined.107 articles were considered suitable for full-text analysis.Articles regarding the use of ABUS in the screening or clinical setting were included. (Lines 33-55)

Point 3: -Line 33-4, this sentence is unclear, please clarify.

Response 3:As a response to Point 3, we modified in the body text the following statements:

Approval became possible after the U-Systems pivotal clinical retrospective multi-reader study. The study included 164 cases (133 non-cancers and 31 biopsy-proven cancers). 17 radiologists first interpreted the mammography images alone, and then interpreted the combined mammography+ABUS. The authors fund that ABUS could detect breast cancer with a clinically insignificant decrease in specificity compared to screening mammography alone (76.2% versus 78.1%; p=0.48). (Lines 33-37)

Point 4:-Please add pics showing benign lesions. In addition, add arrows (Figure 1).

Response 4:As a response to Point 4 we added pictures with benign lesions (fibroadenoma, intramammary lymphnode and angiolipoma) – Figure 10.

We added arrows in Figure 1

Point 5: -Line 98-9, Scanning presure variation, add reference.

Response 5:As a response to Point 5, the reference is (Chang et al. –reference 20) we modified the statement: However, the authors in a recently updated systems, adjusted the compression pressure in five steps with a maximum pressure of 25lbs. (Lines 130-135)

Point 6: -Line 101-102, please redone this sentence since it is unclear.

Response 6:As a response to Point 6 we modified the following statement: The detection rate of breast cancer per 1000 women screen increased with 2.4 [8], 3.9 [6], and even 7.7 [7] for ABUS associated with FFDM compared to FFDM alone. The recall rate per 1000 women screened in the study conducted by Wilczek et al. [8] was 13.8 (95% CI: 9.0, 19.8) for FFDM alone and 22.8 for combined FFDM and ABUS (95% CI: 16.2, 30.0).(Lines 137-141)

Point 7: -Line 128-129, it is important to highlight that HHUS examination will give important information and allow better scanning of the peripheral region.In addition, what about the experience and knowledge of the clinician?

Response 7:An important aspect that would relevant to mention is the fact that HHUS gives valuable information and allows better scanning of the peripheral region. Furthermore, the experience and the knowledge of the clinician increases the quality of the diagnosis. (Lines 185-188)

Point 8: -Line138-139, this statement is unclear for me, Please explain the reason of this asseveration.

Response 8:As a response to Point 8, we’ve made a mistake in inserting the reference, and we modified in the body text the following statement “Some studies have obtained a significantly higher detection rate in ABUS compared to HHUS. Wang et al. [25] reported detection rates of HHUS and ABUS were 95.8% (158/165) and 97.6% (161/165) respectively. Furthermore, the three cases missed by US were found by ABVS, which illustrates a key advantage of an ABUS system for standardised, reproducible and bilateral whole-breast imaging”. (Lines 194-197)

Point 9:-Line 144, this section is confuse since the title do not fit with the information given. Please provide information about this comparison.

Response 9:As a response to Point 9, and we modified in the body text the following statement: “Table 3 shows the comparison in terms of sensitivity and specificity of the ABUS and HHUS in several studies including a significant number of women.

Table 3. Sensitivity and Specificity of ABUS versus HHUS regarding breast cancer detection” (Lines )

Point 10:-Line 226-228: please redone this sentence.

Response 10:As a response to Point 10, we modified in the body text the following statement:  In case that the transducer is not evenly and sufficiently compressed, the air becomes interposed at the edges of the acquired image, hampering the analysis of the glandular parenchyma in the periphery of the image. (Lines 201-204)

Point 11: -Line 247-249: in Figure 5, the skip artifact is unclear to me, Please check. 

Response 11:  As a response to Point 11, we modified in the body text the following statement: “When the transducer quickly slides over a firm and superficial lesion (cyst, fibroadenoma), it creates a linear artifact, which is observed in the coronal plane and sagittal plane. It appears as a horizontal line located superior to the lesion [43]”  (Lines 245-247)

As a response to Point 11, we modified the Figure, which illustrates the “skip artifact”

Point 12:-Posterior enhancement and nipple artifact sections are too short. please extend.

Response 12:As a response to Point 12, we modified in the body text the following statements:

Nipple artifact

The retro areolar region is difficult to assess due to the shadowing artifact induced by the nipple that appears as hypoechoic columns extending in anterior-posterior direction behind the nipple. This artifact can be caused by an imperfect adjustment of irregular nipple surface and if an abnormality is suspected the patient is recalled for rescanning or for a HHUS[43]. (Lines 257-261)

We added a new image illustrating the nipple artifact (Figure 6)

White wall sign         

This artifact is observed with transonic masses (cysts) or some solid masses as in HHUS. The white-wall sign presents as an echogenic wall in the coronal view and corresponds to the acoustic enhancement on HHUS. It appears posterior to the lesion due to less attenuated ultrasounds within the lesion. It can help interpretation, but it is not specific to benign masses due to the fact that high grade carcinomas can also present this artifact (Figure 8) [44-45].

 (Lines 382-388)

We added a new image illustrating the nipple artifact (Figure 8)

Dear reviewer,we would like to thank you for taking the time to carefully evaluate our article. Thank you for your suggestions, appreciation and for your positive response.

Best regards,

Ioana Boca (Bene)

Round 2

Reviewer 1 Report

This was an interesting paper to read. However, there still are some things that could be done better. The english is not perfect and meticoulous proof reading by a native english speaking person or someone equally competent is suggested of the full manuscript.

Address all cost-benefit / economical studies of the technique in the discussion in detail including limitations of studies.

Line 19-21: Omit parenthesis in abstract. Write in new sentence instead.

35: Same as above.

42: instead of doctor write physician or radiologist.

79-85: How many percent of patients was succesfully examined with ABUS and how many could the technique not be applied to?

Author Response

This was an interesting paper to read. However, there still are some things that could be done better.

Point 1:The english is not perfect and meticoulous proof reading by a native english speaking person or someone equally competent is suggested of the full manuscript.

Response to Point 1:The entire manuscript was proofread and once again checked for spelling and grammar errors.

Point 2:Address all cost-benefit / economical studies of the technique in the discussion in detail including limitations of studies.

Response 2: As a response to Point 2 we added the following statement: “To the best of our knowledge this is the only study regarding the economical impact of ABUS utilization in screening. The limitations of the study are that data related to ultrasound recalls rate, process mapping of treatment and care pathway of women affected by cancer were retrieved from real-world data, and the lack of recent information, related to the rate of annual invites to participate in screening programs.” Lines 273-278

Point 3:Line 19-21: Omit parenthesis in abstract. Write in new sentence instead.

Response 3:As a response to Point 3 we modified the following statement: “Many disadvantages can be diminished by additional attention and training. Disadvantages regarding image acquisition are represented by the inability to assess the axilla, the vascularization and the elasticity of a lesion, while concerning the interpretation are artifacts due to poor positioning, lack of contact, motion or lesion related.” Lines 19-21

Point 4:35: Same as above.

Response 4: As a response to Point 4 we modified the following statement: “The study included 164 cases, 133 non-cancers and 31 biopsy-proven cancers.” Line 36

Point 5: 42: instead of doctor write physician or radiologist.

Response 5: As a response to Point 5 replacedthe word “doctor” with physician: “ABUS is a technique that separates the moment of image acquisition (made by the radiographer) from the moment of image interpretation, thus reducing the operator-dependence, as well as the time spent by the physician.” Lines 40-45

Point 6: 79-85: How many percent of patients was succesfully examined with ABUS and how many could the technique not be applied to?

Response 6: As a response to Point 6 we added the following statement: “Breast size is a particular aspect of every woman and variable in the general population. Golatta et al. [11] in their monocentric, exploratory, prospective study that included 983 patients that ≥14.8% had cup D breast size, which may have required additional acquisitions.” Lines 89-93

Dear reviewer,we would like to thank you for taking the time to carefully evaluate our article. Thank you for your suggestions, appreciation and for your positive response.

Best regards,

Ioana Boca (Bene)
